# Association of physical activity intensity and bout length with mortality: An observational study of 79,503 UK Biobank participants

**Louise A. C. Millard**[1,2]*, **Kate Tilling**[1,2], **Tom R. Gaunt**[1,2], **David Carslake**[1,2], **Deborah A. Lawlor**[1,2]

**1** MRC Integrative Epidemiology Unit (IEU) at the University of Bristol, Bristol, United Kingdom, **2** Population health Sciences, Bristol Medical School, University of Bristol, Bristol, United Kingdom

\* louise.millard@bristol.ac.uk

## Abstract

### Background

Spending more time active (and less sedentary) is associated with health benefits such as improved cardiovascular health and lower risk of all-cause mortality. It is unclear whether these associations differ depending on whether time spent sedentary or in moderate-vigorous physical activity (MVPA) is accumulated in long or short bouts. In this study, we used a novel method that accounts for substitution (i.e., more time in MVPA means less time sleeping, in light activity or sedentary) to examine whether length of sedentary and MVPA bouts associates with all-cause mortality.

### Methods and findings

We used data on 79,503 adult participants from the population-based UK Biobank cohort, which recruited participants between 2006 and 2010 (mean age at accelerometer wear 62.1 years [SD = 7.9], 54.5% women; mean length of follow-up 5.1 years [SD = 0.73]). We derived (1) the total time participants spent in activity categories—sleep, sedentary, light activity, and MVPA—on average per day; (2) time spent in sedentary bouts of short (1 to 15 minutes), medium (16 to 40 minutes), and long (41+ minutes) duration; and (3) MVPA bouts of very short (1 to 9 minutes), short (10 to 15 minutes), medium (16 to 40 minutes), and long (41+ minutes) duration. We used Cox proportion hazards regression to estimate the association of spending 10 minutes more average daily time in one activity or bout length category, coupled with 10 minutes less time in another, with all-cause mortality. Those spending more time in MVPA had lower mortality risk, irrespective of whether this replaced time spent sleeping, sedentary, or in light activity, and these associations were of similar magnitude (e.g., hazard ratio [HR] 0.96 [95% CI: 0.94, 0.97; *P* < 0.001] per 10 minutes more MVPA, coupled with 10 minutes less light activity per day). Those spending more time sedentary had higher mortality risk if this replaced light activity (HR 1.02 [95% CI: 1.01, 1.02; *P* < 0.001] per 10 minutes more sedentary time, with 10 minutes less light activity per day) and an even higher risk if this replaced MVPA (HR 1.06 [95% CI: 1.05, 1.08; *P* < 0.001] per 10

these confidential data. Analysis code is available at [https://github.com/MRCIEU/UKBActivityBoutLength/].

**Funding:** This work was supported by the University of Bristol (https://www.bristol.ac.uk/) and UK Medical Research Council (https://mrc.ukri.org/) [grant numbers MC_UU_00011/1, MC_UU_00011/3, MC_UU_00011/4 and MC_UU_00011/6]. LACM is funded by a University of Bristol Vice-Chancellor's Fellowship. The funders had no role in study design, data collection and analysis, decision to publish, or preparation of the manuscript.

**Competing interests:** I have read the journal's policy and the authors of this manuscript have the following competing interests: DAL has received support from Medtronic Ltd and Roche Diagnostics for biomarker research unrelated to this paper. KT has acted as a consultant for CHDI Foundation. TRG receives funding from GlaxoSmithKline and Biogen for unrelated research. Other authors have declared that no competing interests exist.

**Abbreviations:** BMI, body mass index; HR, hazard ratio; MVPA, moderate-vigorous physical activity; PAEE, physical activity estimated energy expenditure; STROBE, STrengthening the Reporting of OBservational studies in Epidemiology; WHO, World Health Organization.

minutes more sedentary time, with 10 minutes less MVPA per day). We found little evidence that mortality risk differed depending on the length of sedentary or MVPA bouts. Key limitations of our study are potential residual confounding, the limited length of follow-up, and use of a select sample of the United Kingdom population.

## Conclusions

We have shown that time spent in MVPA was associated with lower mortality, irrespective of whether it replaced time spent sleeping, sedentary, or in light activity. Time spent sedentary was associated with higher mortality risk, particularly if it replaced MVPA. This emphasises the specific importance of MVPA. Our findings suggest that the impact of MVPA does not differ depending on whether it is obtained from several short bouts or fewer longer bouts, supporting the recent removal of the requirement that MVPA should be accumulated in bouts of 10 minutes or more from the UK and the United States policy. Further studies are needed to investigate causality and explore health outcomes beyond mortality.

## Author summary

### Why was this study done?

- Spending more time active and less time sedentary is associated with many health benefits.

- It is currently not clear whether these benefits differ if time spent active or sedentary is accumulated in longer versus shorter bouts or depending on which type of activity they replace.

- Recent policy changes in the UK and the US removed the recommendation that activity should be accumulated in bouts of 10 minutes of more.

### What did the researchers do and find?

- We used data on 79,503 UK-based adult participants and assessed whether mortality risk differed if people did longer versus shorter bouts of moderate-vigorous physical activity (MVPA) or sedentary behaviour.

- We found that more time spent in MVPA was associated with lower mortality risk irrespective of whether it replaced light activity, sedentary time, or sleep; more time spent sedentary was associated with higher mortality risk, particularly if it replaced MVPA.

- We found little evidence to suggest that mortality risk differs depending on the length of MVPA or sedentary bouts.

**What do these findings mean?**

- Interventions that increase time spent in MVPA or reduce sedentary time (or both) could benefit health.

- Any time spent in MVPA may be beneficial, even bouts of shorter (e.g., 1 minute) duration, supporting recent UK and US policy changes removing the requirement that MVPA should be accumulated in bouts of 10 minutes or more.

## Introduction

Physical activity is associated with many health benefits such as better cardiovascular health and reduced risk of some cancers and type 2 diabetes [1]. A recent systematic review of prospective studies suggested that higher levels of physical activity at any intensity, and less time spent sedentary, are associated with a reduced risk of mortality [2].

Policy in the UK recommends that people accumulate 150 minutes each week in moderate physical activity or 75 minutes in vigorous activity [3], while policy in the US and the World Health Organization (WHO) guidelines have been recently updated to recommend ranges (150 to 300 minutes each week for moderate intensity and 75 to 150 minutes for vigorous intensity) rather than minimum amounts alone [4,5]. Until recently, the advice also stated that activity should be accumulated in bouts of 10 minutes or more, but this has now been removed from the UK, the US, and WHO guidelines [3–5]. These changes were based on evidence from cross-sectional, prospective cohort, and randomised trials. For example, the removal of minimum bout length from WHO guidelines was based on a systematic review [6] of 27 research studies: 13 cross-sectional studies, 2 prospective cohort studies, 1 nonrandomised trial, and 11 randomised trials. The trials had small sample sizes (all ≤255) and short-term follow-up (≤18 months). The largest sample size among the included prospective cohort and cross-sectional studies was 6,321.

Few prospective cohort studies have assessed how the duration of moderate-vigorous physical activity (MVPA) bouts relates to health. A meta-analysis of 29,734 children (4 to 18 years old) across 21 cohort studies found a similar benefit of MVPA on cardiometabolic risk factors across different bout durations [7]. In that study, an isotemporal approach was used to estimate associations of spending more time in one MVPA bout duration category coupled with less time in another MVPA bout duration category. They controlled for the overall time in MVPA to investigate its composition, but did not account for time spent in other activity categories such as sleeping or sedentary [7]. Of 3 studies in adults, 2 found no notable association of MVPA bout length with their respective outcomes: cardiovascular risk factors (N = 2,190) [8] and all-cause mortality (N = 4,840) [9]. The other (N = 3,250) reported smaller mean waist circumference and lower body mass index (BMI) in those who spent more time in MVPA bouts of 10+ minutes rather than shorter bouts [10]. None of these studies considered couplings of activity categories, thus did not examine whether results differed depending on the form of activity substituted for MVPA. They all grouped bouts ≥10 minutes together [7–10]. Other studies have used 2 summary variables to characterise MVPA bouts: (1) the number of bouts; and (2) the average time spent in bouts (in total) per day, but these do not describe the range of bout lengths a person undertakes or how often they undertake them [11–15]. We have found only 1 study that examined the importance of sedentary bout length (N = 7,985

adults) [16]. It found that higher percentage of total sedentary time in shorter sedentary bouts (< = 29 minutes) was associated with lower mortality, but overall time spent sedentary was not accounted for (S1 Fig).

The aim of our study is to examine whether mortality differs depending on time spent in different activity categories (e.g., including sleep and sedentary, not just being physically active) and whether time spent sedentary or in MVPA is accumulated in longer versus shorter bouts. We use a novel analytical approach that addresses limitations of previous studies to assess associations of overall time spent in different activity categories and bout length categories in terms of coupling more time spent in one category with less time in another category.

## Methods

The analysis plan was developed by LACM, DAL, KT, and TRG prior to analyses beginning. We initially sought to investigate the impact of physical activity bout length on BMI, but changed this to all-cause mortality because BMI is measured prior to accelerometer wear in the UK Biobank. Following reviewers' comments, we made one change to our main analysis: splitting our MVPA 1- to 15-minute bout length stratum into 2 categories: 1 to 9 minutes and 10 to 15 minutes, so that the different impact of <10- and > = 10-minute bouts could be directly assessed. We also added 2 additional sensitivity analysis: (1) using the isometric log ratio transformation as an alternative approach to analysing compositional data; and (2) repeating our main analyses excluding the first year and first 2 years of follow-up to explore whether undiagnosed prevalent disease might confound our results. This study is reported as per the Strengthening the Reporting of Observational Studies in Epidemiology (STROBE) guideline (S1 STROBE Checklist).

### Participants

We used data from the UK Biobank participants. UK Biobank is a large prospective cohort of approximately 500,000 adults (5% of those invited [17]) aged 40 to 69 years old at recruitment between 2006 and 2010 [18]. Written informed consent was obtained to collect and store data and bio samples, to link participants to health and administrative data, and for researchers to use these data for health research. UK Biobank received ethical approval from the UK National Health Service's National Research Ethics Service (ref 11/NW/0382). This research was conducted under UK Biobank application number 17810.

In 2013, participants who had provided valid email addresses were invited to participate in the accelerometer substudy, apart from the participants of one assessment centre (3,797 participants; 0.76% of the cohort) who were excluded due to participant burden concerns as they had been invited to participate in pilot studies [19]. Between 2013 and 2015, participants were sent devices in order of acceptance. Of those invited, 106,053 agreed to participate, and 103,711 (44% of invited) provided some accelerometer data [19]. Our study includes 84,176 participants with at least 72 hours accelerometer wear time and no missing data for confounding factors used in our analyses (Fig 1, Section A in S1 Text).

### Data collection

**Physical activity measurement.** Physical activity was measured using the Axivity AX3 wrist-worn tri-axial accelerometer and stored in gravity (g) units. The devices were configured to start recording at 10 AM 2 working days after dispatching to participants by post and to record tri-axial acceleration at a frequency of 100 Hz (dynamic range +/− 8 g) for 7 days. This results in over 181 million values per person (100 values per second across 7 days for each of the 3 axes).

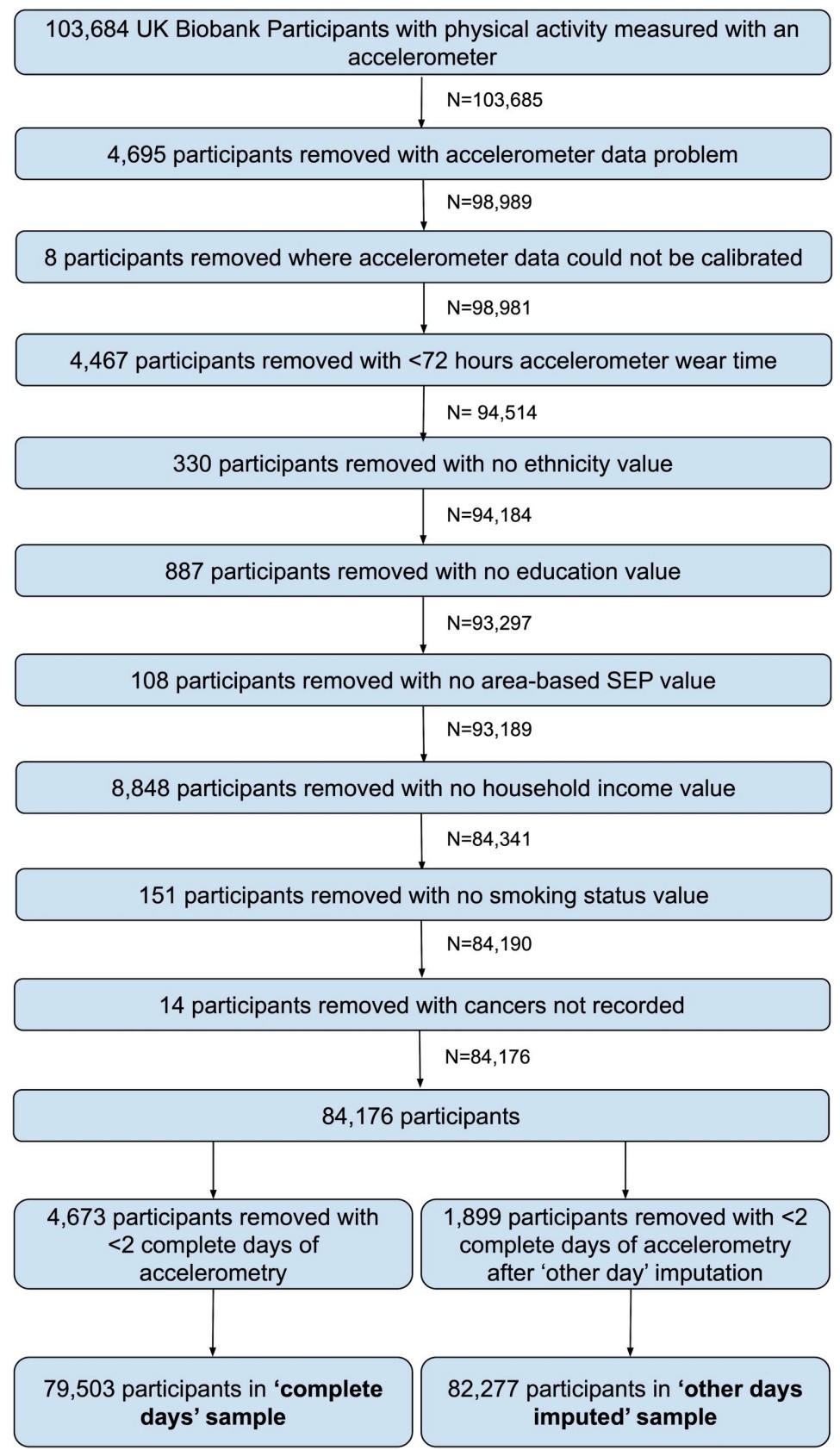

**Fig 1. Participant flow diagram.** SEP, socioeconomic position.

**All-cause mortality.** Date of death was obtained from the NHS Information Centre for participants from England and Wales and the NHS Central Register for participants from Scotland. For each participant, follow-up was defined as starting at the age they stopped wearing the accelerometer. End of follow-up was defined as either participant's age on December 31, 2019 for those in England, Wales, and Scotland (the end of our predefined follow-up period) or their age at death for those who died before the end of follow-up.

**Potential confounders.** We considered the following to be likely confounding factors (based on their known or plausible effects on physical activity and mortality): sex, age at the time of accelerometer wear, ethnicity, socioeconomic position, smoking, BMI, and general ill health (S2 Fig). Sex, ethnicity, and smoking status (never, current, or previous) were self-reported at the baseline assessment. We used education level, household total income, and Townsend deprivation index (a score representing the deprivation of the participant's neighbourhood) to reflect participant's socioeconomic position (see Section B in S1 Text for details of these measures). At the baseline assessment, weight was measured (to the nearest 100 g) in light clothing and unshod using a Tanita BC418MA body composition analyser and height to the nearest cm using a Seca 202 device. We used 3 indicator variables for existing cardiovascular diseases, cancer, and respiratory diseases prior to accelerometer wear as measures of baseline ill health.

The season in which participants wore the accelerometer, while not a confounder since it would not plausibly affect subsequent risk of death, may affect activity. We therefore derived 2 variables denoting the day of the year on which the accelerometer was worn using the cosine function approach, defined as $c_1 = \cos\left(\frac{2\pi d}{365}\right)$ and $c_2 = \sin\left(\frac{2\pi d}{365}\right)$, where $d$ is the day of the year accelerometer wear began [20]. We included $c_1$ and $c_2$ as covariates in our models to reduce the variation in activity exposure variables.

## Accelerometer data preprocessing

We used the UK Biobank accelerometer analysis tool (available at https://github.com/activityMonitoring/biobankAccelerometerAnalysis/) [19,21,22] to preprocess the accelerometer data and derive summary activity variables for each 1-minute epoch in each participant's accelerometer time series. The steps conducted by this tool include resampling x/y/z axes to 100 Hz, calibration to local gravity [23], noise and gravity removal, epoch generation—including both average vector magnitude and machine learning predictions of physical activity categories for each epoch—and nonwear detection. The machine learning model [21] predicts activity categories (sleep, sedentary, walking, light activity, and MVPA) from accelerometer data. It was trained using accelerometer data captured in free-living conditions and labelled with "ground truth" activities from accompanying videos and the Compendium of Physical Activities determined using a body-worn camera [24].

## Statistical analyses

**Dealing with missing accelerometer data.** While the UK Biobank participants were asked to wear the accelerometer continuously for 7 days, 24% of our sample had some missing data. We identified periods of nonwear using the Biobank accelerometer analysis tool, defined as consecutive stationary episodes (where all 3 axes had a standard deviation of less than 13.0 milligravities [m-grav]) lasting for at least 60 minutes (the sensitivity of the accelerometer makes it possible to detect very small movements indicating it is being worn) [19]. We used 2 approaches to explore missing accelerometer data—a "complete days" approach and an "other day" imputation approach—that make different missingness assumptions. The complete days approach uses only days with complete accelerometer data in our analyses (referred to as

"valid" days). The other day imputation approach involved finding all periods of accelerometer data on other days that are during the same time period and have no missing data (including from days with missing data at other times). One of these periods is then randomly chosen as the imputed sequence for the missing region. Imputed "valid" days are those with no missing data after this imputation. Details on missing data assumptions of these approaches are provided in Section C in S1 Text. We report results using the complete days data as our main results, and results using the imputed data are provided in the Supporting information.

**Deriving physical activity bouts.** We assigned each 1-minute epoch (interval) of accelerometer data to an activity category—either sleep, sedentary, light activity, or MVPA. The machine learning model (see "Accelerometer data preprocessing" section above) predicted the activity categories with varying levels of success. For example, while 91% of minutes spent sleeping were correctly classified as sleep, only 25% of light activity minutes were correctly classified as light activity. For this reason, we used a hybrid approach that first identified MVPA as minutes$\geq$100 m-grav (a threshold used in previous research [25]) and then used the machine learning model to identify minutes of sleep and sedentary behaviour from those not already assigned to MVPA [21]. All other minutes not assigned to MVPA, sleep, or sedentary categories were assigned to the light activity category. For each participant, we identified contiguous sequences of 1-minute epochs with a given activity category; these are referred to as activity bouts and can be of any length so long as the participant remains in the same activity category.

As a sensitivity analysis, we used only the machine learning model to define all categories and refer to this as the ML-only approach. As well as categories of MVPA, sleep, and sedentary, this model predicts walking and light activity. There are 2 reasons we used the hybrid approach as our main analysis: (1) the degree of misclassification of the machine learning model for MVPA estimated in the study publishing this model [21] (e.g., only 58% of MVPA minutes were predicted correctly as MVPA); and (2) the activity categorisation in [21] included a separate walking category, whereas we sought to categorise brisk walking as moderate activity and slow walking as light activity, with no separate walking category.

**Deriving summary variables reflecting time spent in activity categories, overall, and in bout length strata.** For both approaches, we calculated, for each participant, the overall time they spent in each activity category, on average per day. Hence, for each participant, the total across all activity categories (for both the hybrid and ML-only approaches) equalled 1,440, the number of minutes in a day. To investigate whether the association of time spent in MVPA (or sedentary) with all-cause mortality changes depending on the time participants spend in bouts of different duration, we categorised them as short (1 to 15 minutes), medium (16 to 40 minutes), and long (41+ minutes). For MVPA, we further split the short category into 2 subcategories (1 to 9 minutes and 10 to 15 minutes) so that bouts <10 minutes and >10 minutes can be compared (at the request of reviewers). We derived the time spent in MVPA and sedentary bouts of each length, on average per day. Further details on our derivation of activity summary variables are provided in Section D in S1 Text.

**Estimating the association of overall time spent in activity categories with all-cause mortality taking account of total time spent in that activity and coupling of spending more time in one activity category with less time in another category.** We used Cox proportional hazards regression to test the association of each activity summary variable with all-cause mortality. All models were performed with age as the time variable. We tested each association before and after adjustment for potential confounders. Exact dates of birth were not available so ages were estimated assuming birth on July 1 in the reported year of birth.

It is possible that BMI and ill health subsequent to activity assessment could mediate the effect of activity on mortality. While we adjusted for BMI and ill health assessed at

baseline (3 to 9 years prior to activity assessment), tracking of these factors across time (e.g., due to factors that affect BMI across the life course) means that BMI and ill health measured before activity are also proxies for these factors measured after activity (S2 Fig). Adjusting for proxies of mediating factors could attenuate our estimates towards the null [26]. We therefore performed a sensitivity analysis excluding BMI and ill health as covariates.

Within 1 day, there are 1,440 minutes so a greater amount of time spent in one activity category must be coupled with a lesser amount of time spent in one or more other activity categories. For this reason, we model associations in terms of couplings of activity categories, in a similar way to our previous activity bigrams approach [27]. We assign, in turn, one activity category as the baseline and estimate the hazard associated with spending 10 minutes less time in this baseline category when coupled with spending 10 minutes more time in a given comparison category, on average per day. Further details of this approach are provided in Section E in S1 Text.

**Relating time spent in short, medium, and long bouts of MVPA or sedentary behaviour with all-cause mortality.** As with our models for overall time, to estimate the association of time spent in sedentary and MVPA bouts of different length with all-cause mortality, we model these associations in terms of couplings of activity categories. Hence, unlike previous work [7–10,16], our models estimate associations for couplings between MVPA and sedentary bouts and also with other activity categories. Further details are provided in Section F in S1 Text.

We conducted sensitivity analyses for all models, starting follow-up 1 year and 2 years after the start of accelerometer wear, to investigate whether our results may be biased by participants reducing their activity due to existing ill health. We also conducted sensitivity analyses using the isometric log ratio transformation, which accounts for the compositional nature of time spent in activity categories [28].

For all models, we generated Schoenfeld residuals and estimated the correlation between log-transformed survival time and the scaled Schoenfeld residuals to test the proportional hazards assumption.

Analyses were performed in R version 3.5.1, Matlab r2015a or Stata version 15, and all of our analysis code are available at https://github.com/MRCIEU/UKBActivityBoutLength/. Git tag v0.2 corresponds to the version of the analyses presented here.

## Results

Of the 84,176 eligible participants, 79,503 and 82,277 were included in our complete days and other day imputed samples, respectively (Fig 1). Other day imputation greatly increased the number of valid days (e.g., 96% and 24% of participants had 7 valid days in the imputed and complete days data; S3 Fig). Descriptive statistics of our main (complete days) sample are provided in Table 1. The total person years at risk in the complete days analysis was 405,438, and 1,615 participants died, giving a mortality rate of 5.10 per 1,000 person years (equivalent numbers for other day imputed sample were 419,520 and 1,688, with a mortality rate of 5.10 per 1,000 person years). Participants who were included in our analyses, compared with those who were invited to wear an accelerometer but did not respond, did not accept or had missing accelerometer of confounder data, were younger, more likely to be white, more educated, and living in an area with less social deprivation, had a higher income, lower BMI, were less likely to have ever smoked, less likely to have a circulatory disease or cancer, were more likely to have worn the accelerometer in winter, and were less likely to die during the follow-up period (S1 Table).

**Table 1. Summary statistics of the UK Biobank participants in our sample.**

| | Mean (SD) or N (%)[a] |
|---|---|
| Age in years at assessment centre (years) | 55.89 (7.83) |
| Sex—% male | 36,196 (45.53) |
| Ethnicity—white | 77,145 (97.03) |
| Black or Black British | 654 (0.82) |
| Asian or Asian British | 686 (0.86) |
| Other | 1,018 (1.28) |
| Smoking status—% ever | 34,270 (43.11) |
| Income (pounds)—less than 18,000 | 11,557 (14.54) |
| 18,000 to 30,999 | 19,197 (24.15) |
| 31,000 to 51,999 | 22,864 (28.76) |
| 52,000 to 100,000 | 20,063 (25.24) |
| >100,000 | 5,822 (7.32) |
| BMI (kg/m$^2$) | 26.71 (4.52) |
| Respiratory disease diagnosis | 32,837 (41.30) |
| Circulatory disease diagnosis | 24,723 (31.10) |
| Cancer diagnosis | 11,157 (14.03) |
| Education—none of the below | 5,931 (7.46) |
| College or university degree | 35,871 (45.12) |
| A levels/AS levels or equivalent | 28,748 (36.16) |
| O levels/GCSEs or equivalent | 42,622 (53.61) |
| CSEs or equivalent | 9,978 (12.55) |
| NVQ or HND or HNC or equivalent | 14,603 (18.37) |
| Other professional qualifications (e.g., nursing and teaching) | 27,986 (35.20) |
| Townsend deprivation index | −1.71 (2.82) |
| Death occurred | 1,615 (2.03) |
| Season—winter (December to February) | 17,372 (21.85) |
| Autumn (September to November) | 23,243 (29.24) |
| Spring (March to May) | 17,697 (22.26) |
| Summer (June to August) | 21,191 (26.65) |

[a] Mean (SD) for continuous and percentage for binary variables. N = 79,503.

A level, Advanced level; AS level, Advanced Subsidiary level; BMI, body mass index; CI, confidence interval; CSE, Certificate of Secondary Education; GCSE, General Certificate of Secondary Education; HND, Higher National Diploma; HNC, Higher National Certificate; NVQ, National Vocational Qualification; SD, standard deviation.

S2 Table shows the distributions of the average number of minutes per day in each activity category. Sedentary time was more often accumulated in bouts of longer duration, and MVPA was more often accumulated in shorter bouts. Participants with more time sedentary on average spent more time in long sedentary bouts and less time in short or medium sedentary bouts (S3 Table). Participants who spent more time in MVPA on average spent more time in all categories of MVPA bout length, but particularly shorter bouts. On average, participants appeared to spend more time in MVPA using the hybrid approach compared with the ML-only approach. Patterns of correlation of overall time sedentary or in MVPA with bout length categories were similar for the hybrid and ML-only approaches. Correlations between the same characteristics derived using the hybrid and ML-only approaches were variable. Overall time sedentary was very strongly correlated (Pearson rho >0.99), with reasonable correlation for the sedentary bout length categories (Pearson rho of 0.73, 0.76, and 0.96 for short, medium,

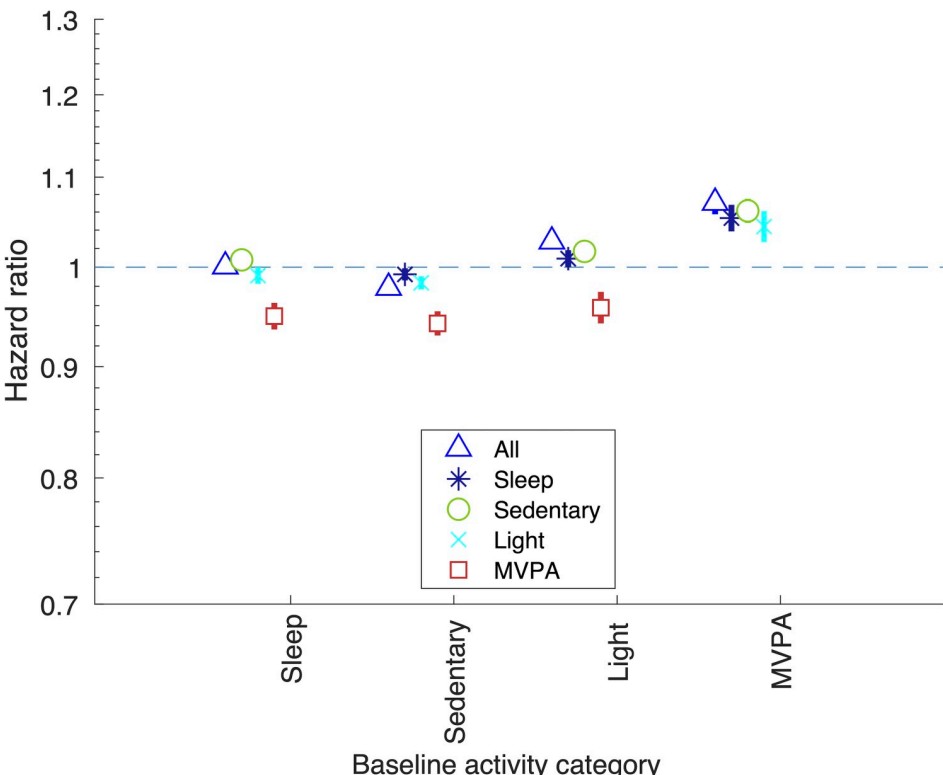

**Fig 2. Associations of less time spent in baseline activity category coupled with more time in comparison category, with all-cause mortality.** HR of spending 10 minutes more time on average per day in comparison activity category, coupled with spending 10 minutes less time in baseline activity category. Using the complete days data. Equivalent results using the other data imputation approach are shown in S6 Fig. Covariates: age at accelerometer wear, sex, ethnicity, season, smoking, SEP (education, Townsend area deprivation index, and income), BMI, and 3 indicators denoting whether the participant had had cardiovascular disease, cancer, or respiratory disease prior to accelerometer wear. Results shown are also provided in S4a Table. BMI, body mass index; HR, hazard ratio; MVPA, moderate-vigorous physical activity; SEP, socioeconomic position.

and long sedentary bouts, respectively) and lower correlations for MVPA (e.g., Pearson rho = 0.45 for overall time spent in MVPA and 0.26 for long MVPA bouts).

## Associations of overall time spent in activity categories, with all-cause mortality

Associations of time spent in activity categories are shown in Fig 2. Overall, time spent in the different activity categories relates differently to mortality. Spending more time in MVPA was associated with lower mortality when coupled with less time spent sleeping, sedentary, or in light activity, and these associations were of a similar magnitude (e.g., hazard ratio [HR] 0.94 [95% CI: 0.93, 0.95; $P < 0.001$] and 0.96 [95% CI: 0.94, 0.97; $P < 0.001$] for 10 minutes more MVPA coupled with 10 minutes less time spent sedentary and in light activity, respectively). Those spending more time sedentary had higher mortality risk if this replaced light activity (HR 1.02 per 10 minutes more sedentary time, with 10 minutes less light activity per day [95% CI: 1.01, 1.02]; $P < 0.001$) and an even higher risk if this replaced MVPA (HR 1.06 per 10 minutes more sedentary time, with 10 minutes less MVPA per day [95% CI: 1.05, 1.08]; $P < 0.001$). Results of sensitivity analyses using the ML-only approach were largely consistent, although there were some differences (e.g., spending more time in light activity coupled with

less time sleeping or sedentary were consistent with the null; S4 Table, S4 Fig). Results attenuated towards the null when starting follow-up 1 year and 2 years after accelerometer wear (S5 Fig).

## Associations of MVPA and sedentary bout length with all-cause mortality

We found little evidence to suggest that associations differed across MVPA bout lengths (Fig 3A, S5 Table). For example, our estimate of association for spending 10 minutes less time in the shortest MVPA bouts (<10-minute duration) coupled with spending 10 minutes more time in long MVPA bouts (40+ minutes duration), with all-cause mortality, was consistent with the null (HR 1.01 [95% CI: 0.93, 1.10; *P* = 0.740]). We also found little evidence that associations differed across sedentary bout lengths (Fig 3B, S6 Table). For example, our estimate of association for spending 10 minutes less time in short sedentary bouts (<16 minutes duration) coupled with spending 10 minutes more time in long sedentary bouts (40+ minutes duration), with all-cause mortality, was consistent with the null (HR 1.03 [95% CI: 0.99, 1.06; *P* = 0.120]). Sensitivity analyses using the ML-only approach showed some differences compared with the hybrid approach (S7 Fig). Most notably, they suggest that spending less time in shorter sedentary bouts coupled with spending more time in longer sedentary bouts, associates with a lower all-cause mortality. Results starting follow-up 1 and 2 years after accelerometer wear were consistent with our main analysis (S8 Fig).

Results of sensitivity analyses using "other day" imputed data were broadly consistent with the results of our main analyses using the complete days data (S4, S6, S7, and S9 Figs, S4–S6 Tables). Results of sensitivity analyses excluding BMI and ill health as covariates were comparable to our main analyses (S4–S6 Tables). Results using isometric log ratio transformed activity variables were consistent with our main analyses (S10–S12 Figs).

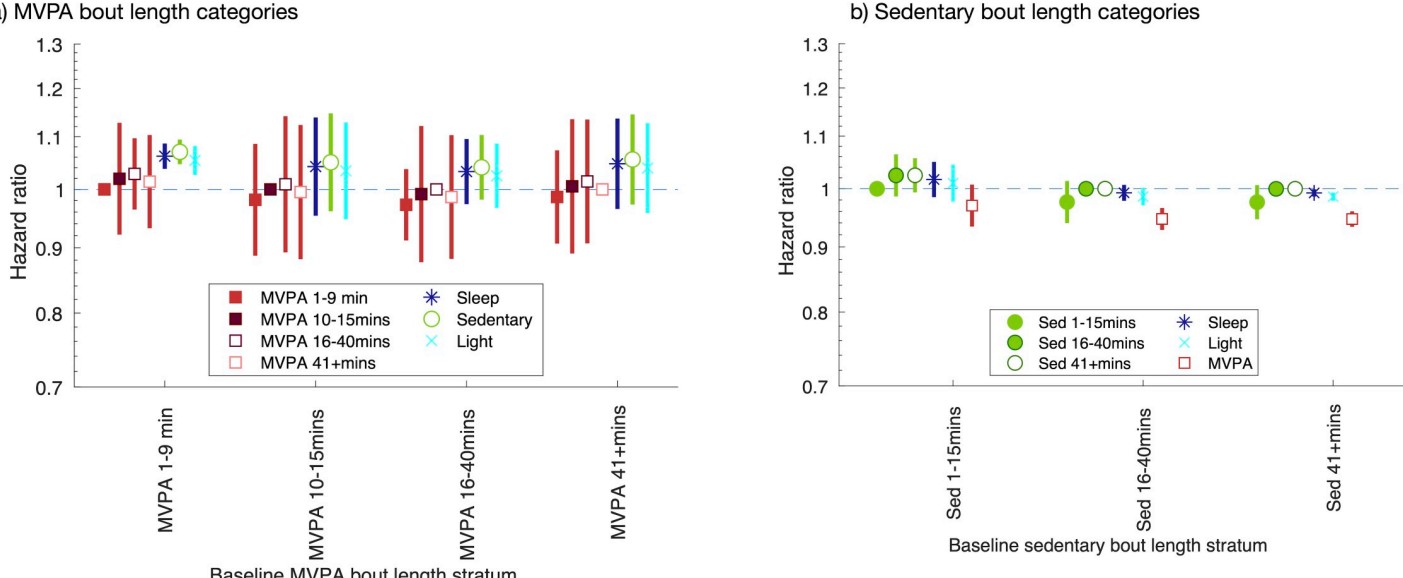

**Fig 3. Associations of time spent in MVPA and sedentary bouts of given duration, with all-cause mortality.** HR of spending 10 minutes more time on average per day in comparison activity category, coupled with spending 10 minutes less time in baseline activity category. Using the complete days data. Equivalent results using "other day" imputation approach are shown in S9 Fig. Covariates: age at accelerometer wear, sex, ethnicity, season, smoking, SEP (education, Townsend deprivation index, and income), BMI, and 3 indicators denoting whether the participant had had cardiovascular disease, cancer, or respiratory disease prior to accelerometer wear. Results shown are also provided in S5a and S6a Tables. BMI, body mass index; HR, hazard ratio; MVPA, moderate-vigorous physical activity; SEP, socioeconomic position.

We found little evidence of violation of the proportional hazards assumption across all Cox regression models (S4–S6 Tables).

## Discussion

In this study, we found that time spent in MVPA was associated with lower mortality, irrespective of whether it was coupled with less time spent sleeping, sedentary, or in light activity and irrespective of whether it was obtained from several short bouts or fewer longer bouts. We also found that time spent sedentary was associated with higher mortality if it was coupled with less time in light activity (but to a lesser extent than if it was coupled with less time in MVPA). These findings emphasise the specific importance of MVPA. They also support recent changes to policy in the UK and the US and WHO guidelines that have removed the suggestion that MVPA should be accumulated in bouts of at least 10 minutes [3–5]. Those policy changes were made on the basis of cross-sectional, prospective cohort, and randomised controlled trial evidence, but those studies were small (e.g., in the systematic review on which this change in WHO guidelines was based, the largest observational study had 6,321 participants and the largest trial had 255 participants [6]).Our results do not support the specific promotion of accumulating MVPA in several smaller bouts but rather suggest that accumulating MVPA in any bout length could reduce risk of premature mortality. Similarly, they also suggest that replacing sedentary periods of any length with light activity, and, to a greater extent, with MVPA, could be beneficial. This is an important public health message as it allows people with different preferences and lifestyles to improve health through accumulating activity in different ways.

Importantly, the methods that we have used here address limitations of other studies that appear not to have controlled for overall time spent across all bout lengths of a given activity category [16], considered that greater amounts of one activity should be coupled with lesser amounts of another [8,10,16] or assessed each coupling combination [7,8,10,16]. We provide all of our code (https://github.com/MRCIEU/UKBActivityBoutLength/) so that others can use this method for exploring other outcomes, or risk factors for different patterns of activity, and examine associations in other studies with similar accelerometer data.

To our knowledge, there is only one existing study that assessed the association of MVPA bout length with mortality; it was considerably smaller than our study ($N = 4,840$), and, consistent with our findings, found no strong evidence of association between MVPA bout length and mortality [9]. Our findings contrast with those of a previous study that analysed sedentary bout length and concluded that longer versus shorter sedentary bouts (defined on the basis of the percent of all time spent sedentary) were associated with a higher risk of premature mortality [16]. We hypothesise that their results may be explained by an effect of total time spent sedentary on all-cause mortality, which was not taken into account in that study.

### Study strengths and limitations

Strengths of this study include the large sample size and use of accelerometer data rather than self-report to measure activity and the prospective nature of the study. We have developed and used a method that appropriately accounts for coupling of activities. We have appropriately explored associations of total time spent in MVPA and sedentary with mortality, including whether this differs by bout length and depending on what alternative activity it is coupled with. This was possible because of our use of accelerometer data and would not be possible using the UK Biobank self-reported activity data. The UK Biobank self-reported data (or most other self-reported data) on activity bouts cannot be analysed in a compositional way because they do not include time spent in bouts of different length of each activity category (only

average time spent in bouts for some activities or the number of days the participant did at least 10 minutes of moderate or vigorous activity). We undertook sensitivity analyses to assess missing accelerometer data assumptions. The code for generating our variables is freely available so can be used by others to explore associations with other health outcomes in the UK Biobank and in other studies with similar activity data.

Our study has a number of limitations. We used a previously published machine learning model to predict activity categories, and so it is possible that misclassifications of those predictions biased our estimates of association. For example, the model uses some orientation specific movement variables, and it is possible that the accelerometer orientation varied between participants. However, our main analysis used a hybrid approach where MVPA was identified using a threshold (>100 m-grav), since prediction accuracy for MVPA from the machine learning model was particularly low. This also has the benefit that average activity (denoted using the average vector magnitude) used to define MVPA in our hybrid approach is orientation independent. We also conducted sensitivity analyses using the machine learning predictions only (ML-only). These results were largely consistent for associations with overall time spent in each activity category, but showed some differences for our bout length results that may be due to biases in the types of activities assigned as MVPA by the ML-only approach compared with the hybrid approach. Further work is needed to compare the types of misclassifications of the hybrid and ML-only approaches.

Participants tended to spend relatively little time in MVPA overall and have MVPA bouts of short duration (the most common bout length was 1 minute, which was the shortest possible bout length in our data) so these estimates were imprecise. Further studies are needed in larger samples (e.g., when larger cohort studies are created) and with more precise measures of MVPA activity bouts (e.g., through more accurate prediction of MVPA using machine learning) to further explore these associations. We chose to use the same bout length strata for MVPA and sedentary behaviour for consistency, but we may have had more statistical power by defining strata according to the distribution of bout lengths for each category (e.g., participants spent more time in longer (versus shorter) sedentary bouts and more time in shorter (versus longer) MVPA bouts). We used 1-minute epochs to derive activity bouts (e.g., a 10-minute bout is a set of 10 adjacent 1-minute bouts), but using a different epoch definition may affect the values of derived bout variables and hence our results [29].

While we accounted for known, measured confounders, our analyses may be biased by residual confounding. It is possible that adjustment for other confounders might attenuate results (e.g., of overall time spent in MVPA) to the null. For example, it is possible that having mobility limitations, or little access to green space or facilities to be physically active, might be related to less time spent in light activity or MVPA and more sedentary behaviour and also to increased risk of mortality during follow-up. Adjustment for 3 different measures of socioeconomic position, including an area-based measure and BMI, is likely to have controlled for some of the potential confounding by these and therefore potentially reduced residual confounding [30]. Residual confounding could also occur due to undiagnosed underlying chronic disease, which could result in being less active and more sedentary, and be associated with increased mortality, particularly in the early years of follow-up. To explore this, we conducted sensitivity analyses starting follow-up 1 and 2 years after accelerometer wear. Results from these analyses showed some attenuation towards the null for our overall time spent in activity categories, which may suggest that our results are biased by confounding with existing ill health, but might also be explained by any true effect of activity on mortality being short term. Longer follow-up time would allow further sensitivity analyses starting follow-up 5 years after accelerometer wear. This, and repeat assessments of physical activity, would help to ensure

that associations are not due to confounding via existing ill health and to explore the impact of changes in activity levels and whether any beneficial effect of activity might be short term.

Our use of time spent in each activity category and in activity bout length strata does not account for variability of activity levels within each of these. For example, participants spending more time in short MVPA bouts may have higher activity intensity levels within these compared to those spending more time in longer MVPA bouts. It also does not account for energy expenditure. Other recent work assessing the association of physical activity estimated energy expenditure (PAEE) with mortality, also in the UK Biobank, found that higher overall PAEE was associated with lower mortality and that associations were stronger with an increasing time spent in MVPA [31].

UK Biobank is a highly selected sample of the UK population with a response rate of 5.5% [32], and evidence suggests that those who volunteered are more affluent and healthy than those who did not [17]. The participants who were included here were also a more affluent and healthier group than the UK Biobank participants who were not included. This "selection" may mean our estimates are biased (see Section G in S1 Text for further discussion of this). Most of the participants in the UK Biobank are of white European origin, and our results may not generalise to other populations.

To conclude, we have used a novel approach to assess whether time spent in different activity types, and in short, medium, or long bouts of MVPA and sedentary behaviour, are associated with all-cause mortality. Our study confirms a strong association between active time and lower mortality, particularly for MVPA compared with light activity. We found little evidence that associations with time spent in MVPA or sedentary differ according to bout length. These results support the recent decision to amend the UK and the US physical activity guidelines to remove the advice that MVPA should be accumulated in bouts of 10 minutes or more [3,4]. Further work is needed to replicate our results in independent data and to investigate causality. Finally, our results highlight the importance of the isotemporal "coupling of time" perspective and suggest that this should be commonplace in any activity analyses, as public health advice based on increasing time spent in a given activity type is misleading without accompanying details of the activities from which this time should be taken.

## Supporting information

**S1 STROBE Checklist. STROBE, STrengthening the Reporting of OBservational studies in Epidemiology.**
(DOC)

**S1 Fig. Directed acyclic graph illustrating path between bout length and survival through a common cause of bout length and total time spent sedentary.**
(PDF)

**S2 Fig. Directed acyclic graph illustrating hypothesised confounding factors and potential mediators.**
(PDF)

**S3 Fig. Distributions of the number of valid days of accelerometer data for participants included in our samples.**
(PDF)

**S4 Fig. Association of lower amounts of time spent in baseline activity category coupled with higher amounts of time in comparison category, with all-cause mortality using "other**

day" imputed accelerometer data.
(PDF)

**S5 Fig. Results of sensitivity analysis using activity predictions only: Associations of lower amounts of time spent in baseline activity category coupled with higher amounts of time in comparison category, with all-cause mortality.**
(PDF)

**S6 Fig. Results of sensitivity analysis starting follow-up 1 and 2 years after accelerometer wear.**
(PDF)

**S7 Fig. Results of sensitivity analysis using ML-only approach: Association of time spent in MVPA and sedentary bouts of a given length, with all-cause mortality.** MVPA, moderate-vigorous physical activity.
(PDF)

**S8 Fig. Results of sensitivity analysis starting follow-up 1 and 2 years after accelerometer wear: Association of time spent in MVPA and sedentary bouts of a given length, with all-cause mortality.** MVPA, moderate-vigorous physical activity.
(PDF)

**S9 Fig. Association of time spent in MVPA and sedentary bouts of a given length, with all-cause mortality using hybrid approach and "other day" imputed data.** MVPA, moderate-vigorous physical activity.
(PDF)

**S10 Fig. Results of sensitivity analysis using isometric log ratio transformed activity variables: Association of time spent in activity categories, with all-cause mortality.**
(PDF)

**S11 Fig. Results of sensitivity analysis using isometric log ratio transformed activity variables: Association of time spent in MVPA bouts of a given length, with all-cause mortality.** MVPA, moderate-vigorous physical activity.
(PDF)

**S12 Fig. Results of sensitivity analysis using isometric log ratio transformed activity variables: Association of time spent in sedentary bouts of a given length, with all-cause mortality.**
(PDF)

**S13 Fig. Directed acyclic graph illustrating the potential for collider bias due to selection into the study sample.**
(PDF)

**S1 Table. Summary statistics of the UK Biobank participants who are in our sample, compared with those who were invited to wear an accelerometer but are not in our sample, for complete days version.**
(DOCX)

**S2 Table. Summary of time spent in activity classifications on average per day.**
(DOCX)

**S3 Table. Correlations between activity summary variables of main analysis (hybrid approach) and ML-only sensitivity approach.**
(DOCX)

**S4 Table. Associations of transferring time between overall activity categories, with all-cause mortality.**
(DOCX)

**S5 Table. Associations of transferring time from MVPA bout length category to other activity category, with all-cause mortality.** MVPA, moderate-vigorous physical activity.
(DOCX)

**S6 Table. Associations of transferring time from sedentary bout length categories to other activity category, with all-cause mortality.**
(DOCX)

**S1 Text. Section A:** Description of participant flow. **Section B:** Potential confounders. **Section C:** Missing data assumptions and imputation of accelerometer data. **Section D:** Deriving activity summary variables. **Section E:** Estimating the association of less time in a given activity category, when coupled with more time in another category. **Section F:** Estimating the association of less time in a given MVPA bout length stratum, when coupled with more time in another activity category/MVPA bout length stratum. **Section G:** Possible bias due to conditioning on a collider. MVPA, moderate-vigorous physical activity.
(DOCX)

## Acknowledgments

This research has been conducted using the UK Biobank Resource.

## Author Contributions

**Conceptualization:** Louise A. C. Millard.

**Data curation:** Louise A. C. Millard.

**Formal analysis:** Louise A. C. Millard.

**Investigation:** Louise A. C. Millard, Kate Tilling, Tom R. Gaunt, David Carslake, Deborah A. Lawlor.

**Methodology:** Louise A. C. Millard, Kate Tilling, Tom R. Gaunt, David Carslake, Deborah A. Lawlor.

**Software:** Louise A. C. Millard.

**Writing – original draft:** Louise A. C. Millard.

**Writing – review & editing:** Louise A. C. Millard, Kate Tilling, Tom R. Gaunt, David Carslake, Deborah A. Lawlor.

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
